# Maternal and neonatal canine cortisol measurement in multiple matrices during the perinatal period: A pilot study

**Debora Groppetti**[1], **Sara Meazzi**[1], **Joel F. S. Filipe**[1], **Carla Colombani**[2], **Sara Panseri**[3], **Sergio A. Zanzani**[1], **Clara Palestrini**[1], **Simona Cannas**[1], **Alessia Giordano**[1]*, **Alessandro Pecile**[1]

**1** Department of Veterinary Medicine, Università degli Studi di Milano, Milan, Italy, **2** Department of Agricultural and Environmental Sciences—Production, Landscape, Agroenergy, Università degli Studi di Milano, Milan, Italy, **3** Department of Veterinary Science for Health, Animal Production and Food Safety, Università degli Studi di Milano, Milan, Italy

* alessia.giordano@unimi.it

## Abstract

Stress exposure during perinatal period may lead to maternal cortisol increase that negatively affects the offspring development. In recent years, the interest on non-invasive sampling methods to measure cortisol as a marker of stress is increasing in both humans and animals. Indeed, discomfort due to blood collection may compromise the diagnostic outcome, mainly in uncooperative patients. So far, some alternative matrices but not milk have been explored in adult dogs, while no data are available on the neonate and paediatric live pups. This study aimed to measure cortisol concentration in different biological substrates in both dams (blood, saliva, hair and milk) and pups (saliva and hair) at established times from proestrus up to two months after parturition. For this purpose, five female German shepherd bitches and their 22 pups were enrolled. Cortisol concentration was assessed using the enzyme immunoassay kit (Salivary Cortisol ELISA kit, Salimetrics) after matrices appropriate preparation if required. Cortisol was measurable in all the substrates, except some milk samples below the detection limit. Maternal cortisol concentrations differed among the matrices ($P$ <0.0001) with the highest values recorded in plasma (median 0.596 µg/dL) compared to saliva (median 0.159 µg/dL), hair (median 0.083 µg/dL) and milk (median 0.045 µg/dL). Cortisol in dams did not vary within the same matrix over time. In pups, salivary (median 0.295 µg/dL) cortisol was always higher than hair (median 0.049 µg/dL; $P$ <0.0001). At birth ($P$ = 0.01) and two months later ($P$ = 0.05), neonatal salivary cortisol was higher compared to other samplings. The present study demonstrates the suitability of these innovative substrates for cortisol measurement, suggesting them as potential diagnostic support in canine neonatology and welfare.

**Data Availability Statement:** All relevant data are within the manuscript and its S1, S2 Tables.

**Funding:** This work was supported by the Università degli Studi di Milano [Linea2_2015_DG]. The funders had no role in study design, data collection and analysis, decision to publish, or preparation of the manuscript.

**Competing interests:** The authors have declared that no competing interests exist.

## Introduction

Pregnancy and the first months of life are critical periods of mammal development [1, 2]. An unsuitable environment that exposes the mother to stress can lead to hypothalamic-pituitary-adrenal axis stimulation ending in a cortisol increase [3, 4]. In turn, due to its possible placental crossing [5], high persistent maternal cortisol concentration may result in intrauterine growth restriction, premature delivery, low birth weight, and compromise neurodevelopment, behaviour and immune function of the offspring [6–8]. Even after birth neonates are fragile and very vulnerable to stressful events, that may increase heart rate and blood pressure while decreasing oxygen levels [9]. Despite cortisol is a well-known stress marker in adult dogs [10], its role as an early predictor in pups has not been established yet. To date, cortisol has been measured in different biological substrates other than blood, such as milk [8, 11], saliva [12], urine [13], faeces [14], and hair [15, 16] in both humans and animals. Indeed, non-invasive sampling procedures are increasing in attractiveness because of ethical reasons and to avoid the fear or physical restraint cortisol rising [8]. Moreover, blood sampling in pups may fail to collect an adequate volume for the analysis. Acute stress can be inferred by measuring cortisol concentration in biological fluids, while hair cortisol mainly provides information on long-term exposure to chronic stress [16, 17]. Particularly concerning milk, in dairy cows the measurement of milk cortisol has been suggested as a biomarker of adverse environmental or pathophysiological conditions [18]. Moreover, literature suggests a link between cortisol concentration and milk composition in women. Indeed, especially in behavioural studies, an evaluation of milk energy together with cortisol measurement is recommended [2].

So far, cortisol has been measured in plasma [19], saliva and hair of adult dogs but no data are available about live canine neonatal and paediatric patients (especially those younger than two weeks), nor about milk cortisol in dogs. This study was designed to assess the hypothesis that canine cortisol may be measured from biological substrates other than blood, making its collection feasible in neonate patients without discomfort. Results will provide preliminary data that could be useful as a starting point for further investigations. A cortisol raise is expected in response to pain, illness or unsuitable environment. However, those conditions are hardly recognizable in the canine neonatal patients. More specifically, the main aim of the present study was to measure the canine cortisol concentration in maternal blood as well as in saliva, hair and milk, and in neonatal saliva and hair at fixed times from proestrus to two months after parturition. Cortisol differences among matrices and sampling times in dams and their pups were also evaluated, in order to highlight a possible substrate that could replace blood sampling. Furthermore, possible correlations between maternal and neonatal cortisol concentrations, and some clinical aspects in German shepherd dogs were investigated. Finally, due to the suggested link between milk composition and cortisol [2], the analysis of milk component was performed.

## Materials and methods

### Animals

The present study was conducted under the approval of the Ethical Committee of the Università degli Studi di Milano (protocol OPBA_106_2017). After the informed consent was obtained from the owner, five healthy purebred female German shepherd dogs were enrolled. The enrolment of dogs living in the same breeding facility and belonging to the same breed was performed in order to reduce possible confounding factors. A physical examination was performed for each dog by a veterinarian specialized in reproduction (DG). All dogs were housed in the same professional breeding facility, provided with both outdoor and indoor

area, off-leash walking zone. They were fed with the same dry commercial diet and exposed to the same environmental conditions. Human interactions were ensured both in dams and pups for at least four hours a day, about two hours in the morning and two hours in the evening, while co-specific contact was reserved only to the mothers before whelping and after weaning. Only dams delivering by vaginal eutocic parturition were included. Reproductive cycle from proestrus to parturition was monitored as previously described [20]. Maternal age, body weight, BCS (scale 1–5), parity, duration of pregnancy and delivery were recorded as well as litter size, pups gender, neonatal body weight and presence of malformations, neonatal morbidity and mortality until two months of life.

## Biological samples collection

The sample collection was performed between 8:00 and 12:00 AM at the kennel (which was located close to the laboratory) at established times as shown in Table 1.

Saliva, blood, colostrum/milk and hair samples were withdrawn in this order. All the samplings were performed by the breeder, who is also a veterinarian.

Blood samples (1.5 mL) were collected from the cephalic vein into K2EDTA tubes and immediately centrifuged at 1500 x g for 10 min to obtain plasma, which was stored at -80˚C.

At least 30 min away from the meal, saliva (min 25μL) was collected with the Salivette sampling device (Sarstedt), following manufacturer instruction. Briefly, the synthetic swab provided with the device was kept in the dog mouth for about one minute and then placed in the test tube. The device was then centrifuged at 1000 x g for two min in order to allow complete saliva retrieval. The samples were stored at -80˚C until further analysis.

Nipples were gently manual milked, and the colostrum/milk sample obtained was divided into two aliquots of at least 5 mL each one for both cortisol and milk composition analysis (see below). No more than 10 mL of colostrum/milk was taken in order not to compromise the amount available for pups. Samples for milk composition were stored at -80˚C, whereas milk aliquot for cortisol measurement underwent extraction procedures. Specifically, these samples were centrifuged at 4000 x g for 20 min and fat was removed from the surface using a vacuum pump. The skimmed samples were stored at -80˚C.

Hair samples were obtained by shaving close to the skin with a clipper an area of about 1 cm2 from the ventral surface of the tail base (8 mg was the minimum weight required for the analysis). Maternal hair was shaved three times at about two-month interval, as shorter collection time resulted in an insufficient amount of regrowth hair. In pups, the rate of hair growth was faster allowing three samplings at about 20 and 40-day time distance. Due to previous conflicting reports about cortisol concentration depending on hair colour [21], only light hair was collected, then stored in a white paper envelope and kept in a -80˚C until the analyses.

Table 1. Collection times of matrices in dams and pups.

| | Sampling time | Matrices | |
|---|---|---|---|
| | | Dams | Pups |
| T0 | The fifth day of proestrus | blood, saliva, hair | None |
| T1 | The twenty-fourth day of pregnancy (at ultrasound confirmation) | blood and saliva | None |
| T2 | The parturition day | blood, saliva, colostrum, hair | saliva and hair |
| T3 | Twenty-one days after parturition (right before the weaning period) | blood, saliva, milk | saliva and hair |
| T4 | Sixty days after parturition | blood, saliva, hair | saliva and hair |

### Cortisol assay

Plasma and saliva were analysed without any further preparation. Hair cortisol extraction was performed according to methods reported in the literature [22] and the dried extract was resuspended with 25 µL of phosphate buffer. Colostrum/milk cortisol extraction was performed by mixing a volume of 100 µL of skim milk with 900 µL ethyl ether and shaken. The supernatant was moved into a tube and dried using a centrifugal evaporator. The residue was dissolved in 125 µL of PBS and vortexed [23]. Cortisol concentrations in all the matrices were assessed using the enzyme immunoassay kit (Salivary Cortisol ELISA kit, Salimetrics) (intra-assay CV = 4,6%; inter-assay CV = 4%).

### Milk composition analysis

The fat percentage was determined as described by Collares et al. [24]. The lactose content was determined following the modified methods by Aliani and Farmer [25] and Fang et al. [26]. Milk energy was calculated as kCal and kJ, determined by the sum of protein, fats and carbohydrates multiplied by the conversion factors related to the macronutrients.

### Statistical analysis

In bitches, the comparison among cortisol concentrations in each matrix at different times was performed by non-parametric statistical Friedman and Wilcoxon's tests. A Kruskal-Wallis followed by Wilcoxon's test was used to compare cortisol concentrations at the same times among different matrices. The correlation between maternal cortisol concentrations and clinical data (maternal age and body weight, BCS, parity, duration of pregnancy and delivery, milk composition, litter size) was obtained using Spearman's rho correlation. In pups, two Generalized Linear Mixed Models (GLMMs) were implemented to determine the significant predictors of salivary and hair cortisol. Logarithmic transformed individual salivary and hair cortisol were introduced as the dependent variables. Maternal age, sampling time, duration of pregnancy, litter size, pups gender, neonatal body weight and malformations, and maternal salivary and hair cortisol were introduced in the GLMMs as independent variables. The identity of each pup and litter were included as random intercept effects. The final models were chosen by backward elimination and best-corrected Akaike information criterion (AICC). Statistical analyses were performed using the Analyse-it software, version 2.21 (Analyse-it Software Ltd, Leeds, UK) and SPSS ver. 20.0 (IBM, Chicago, IL). Statistical significance was set at $P < 0.05$.

## Results

The results reported in the text are expressed as mean ± standard deviation. Maternal age ranged 2 to 8 years (4 ± 3.5). At T0 body weight was 25.50 to 32 kg (27.25 ± 2), at T1, T2, T3 and T4 was 29.4 ± 3.3 kg, 30.6 ± 4.5 kg, 30.6 ± 4.3 kg and 28 ± 2.6 kg, respectively. BCS varied from 2.6 ± 0.5 at T0 to 3 ± 0.71 at T1, 2.8 ± 0.5 at T2, 3 ± 0.7 at T3 and 2.8 ± 0.8 at T4. Two bitches were nulliparous and the remaining three dogs did one to four previous parturitions. All the deliveries occurred between 65 and 68 days (66.2 ± 1.1) from the estimated LH surge. Duration of delivery ranged 2 to 9 h (4.8 ± 3.4). A total of 22 pups all af them born alive were delivered. Litter size ranged from two to six pups (4.4 ±1.8). Fourteen pups were male and eight female. Two pups showed severe congenital defects, namely a severe cleft palate (ID. 2.2) and a mega oesophagus (ID. 1.3). They were humanly euthanized 24 hours and 2 months after birth, respectively. Therefore, neonatal mortality was 9.1%. Birthweight ranged 432 to 1140 gr (698.2 ± 21.5) then increased 740 to 2650 gr (2058 ± 162.8) and 2060 to 11000 gr

**Table 2. Age, parity, litter size body weight and body condition score of the dams enrolled in the study.**

| ID. | Age$_y$ | Parity | N$_p$ | T0 | | T1 | | T2 | | T3 | | T4 | |
|---|---|---|---|---|---|---|---|---|---|---|---|---|---|
| | | | | BW | BCS | BW | BCS | BW | BCS | BW | BCS | BW | BCS |
| 1 | 5.5 | 3 | 6 | 25.5 | 3 | 29.0 | 4 | 28.0 | 3 | 25.5 | 3 | 25.2 | 3 |
| 2 | 8 | 4 | 2 | 26.0 | 3 | 25.9 | 3 | 25.9 | 3 | 28.6 | 4 | 28.4 | 4 |
| 3 | 4.5 | 1 | 6 | 30.0 | 2 | 30.3 | 2 | 32.0 | 2 | 34.0 | 3 | 28.0 | 2 |
| 4 | 2 | 0 | 3 | 32.0 | 2 | 34.5 | 3 | 37.5 | 3 | 36.0 | 3 | 32.0 | 2 |
| 5 | 2 | 0 | 5 | 27.5 | 3 | 27.1 | 3 | 29.5 | 3 | 29.0 | 2 | 26.5 | 3 |

Age$_y$: age expressed in years; N$_p$: number of pups at birth; BW: body weight (kg); BCS: body condition score (1–5 scale).

(7580 ± 715.5) at T3 and T4, respectively. All the data about dams and delivery are reported in Table 2.

## Cortisol measurement in dams

Cortisol was measurable in all the matrices analysed at each collection time (Table 3), except in milk samples of two dams at T2 (ID. 4 and 5) and three dams at T3 (ID. 3, 4, 5) due to a cortisol concentrations below the detection limit. For this reason, milk cortisol values were excluded from statistical comparisons. A dam (ID. 4) got involved in a minor car accident at T4, just before the sampling collection. Its plasma, saliva, and hair cortisol concentrations at T4 were 3.81 µg/dL, 1.41 µg/dL, and 0.057 µg/dL, respectively. Maternal cortisol concentrations differed among the matrices in the entire caseload, with plasma cortisol that resulted always the highest, followed by saliva and hair ($P < 0.0001$). The comparison among the matrices at a specific sampling time provided significant results as well (Fig 1). Specifically, cortisol was significantly higher in plasma than in hair at T0 ($P = 0.04$) and T2 ($P = 0.02$), but not at T4. Compared to saliva, plasma cortisol resulted significantly higher only at T2 ($P = 0.02$). Cortisol concentration did not vary within the same matrix over time.

Composition of colostrum collected at parturition (T2) was: 128.5 ± 49.5 kCal, 538.1 ± 207 kJ, 6.3 ± 0.5 mg/mL protein, 10.7 ± 5% fat, 6.6 ±2.3% carbohydrates (lactose). Milk composition at T3 was: 134.7 ± 18.3 kCal, 564.1 ± 76.6 kJ, 6.1 ± 0.2 mg/mL protein, 11.4 ± 2.1% fat, 7.9 ±2.6% carbohydrates (lactose). The minimum amount of colostrum/milk required to perform analysis was assessed at 3 mL.

## Cortisol measurement in pups

Cortisol values obtained in saliva and hair of pups are reported in Table 3 as well. Cortisol concentration significantly differed between the two matrices in the entire caseload, with saliva cortisol always higher than in hair ($P < 0.0001$). The independent variable of the first

**Table 3. Median, I-III interquartile ranges, minimum and maximum values of cortisol concentration (µg/dL) in dams and pups in different matrices on the entire caseload.**

| | Dams | | | | Pups | | | |
|---|---|---|---|---|---|---|---|---|
| | n | Median | IQR I—III | Min–Max | n | Median | IQR I—III | Min–Max |
| Plasma | 22 | 0.596 | 0.270–0.936 | 0.043–3.814 | - | - | - | - |
| Saliva | 24 | 0.159 | 0.110–0.264 | 0.048–1.410 | 63 | 0.295 | 0.182–0.508 | 0.000–5.682 |
| Hair | 17 | 0.083 | 0.024–0.101 | 0.000–0.177 | 63 | 0.049 | 0.008–0.127 | 0.000–0.720 |
| Colostrum/Milk | 5 | 0.045 | 0.035–0.058 | 0.034–0.061 | - | - | - | - |

n: total number of analysed samples; IQR: interquartile range.

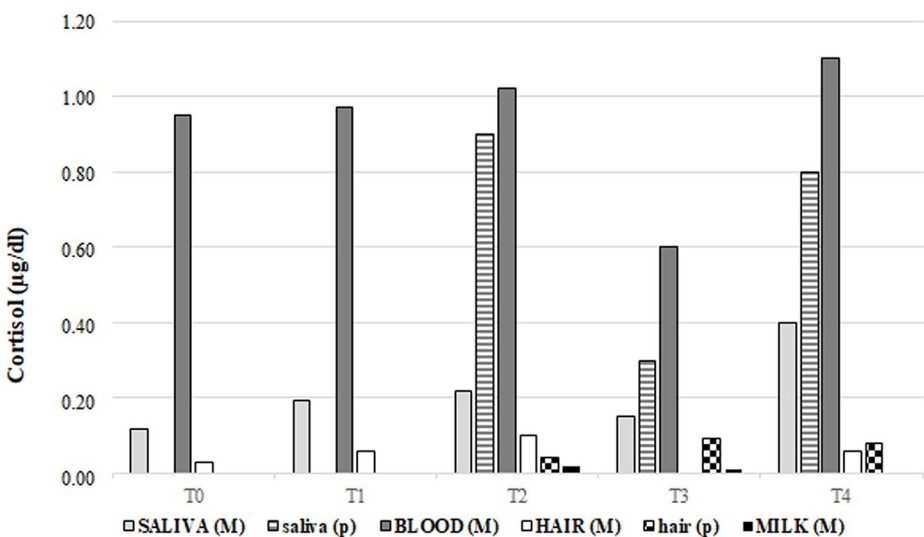

**Fig 1. Maternal and neonatal cortisol concentrations in different matrices over time.** T0: the fifth day of proestrus; T1: the twenty-fourth day of pregnancy; T2: the parturition day; T3: twenty-one days after parturition; T4: two months after parturition; M = maternal; p = pups.

implemented GLMM was salivary cortisol; the time of sampling was the only significant predictor in the final model. No significant predictors were found for hair cortisol in the second GLMM. When continuous predictors were fixed at their mean values, pairwise comparisons showed a significant difference in logarithmic transformed salivary cortisol in pups at T2 and T4 respect to T3 ($P < 0.01$ and $P < 0.05$, respectively; Fig 2). No significant difference was found between T2 and T4 ($P = 0.696$).

## Cortisol correlation with clinical features

Maternal age, body weight and BCS did not affect maternal cortisol concentrations. Nevertheless, bitches who lost weight between 21 days (T3) and 60 days (T4) after parturition showed

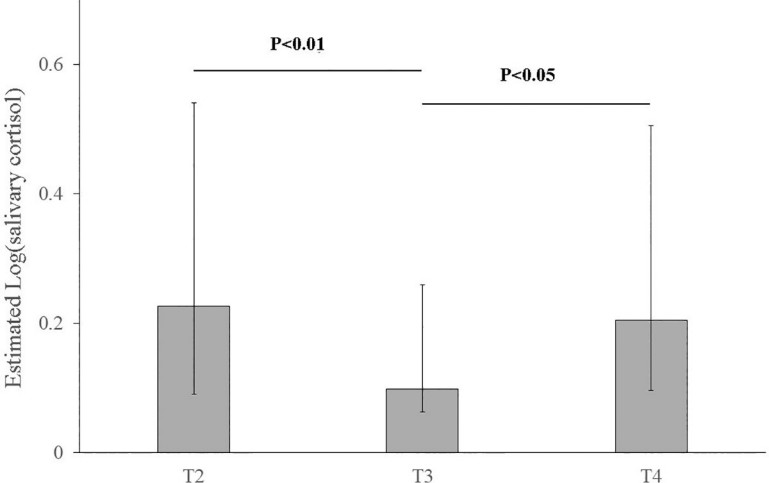

**Fig 2. Logarithmic transformed salivary cortisol estimated by the model (grey columns) in the German shepherd pups.** Horizontal black bars: significance of pairwise comparisons between different times of sampling (T2 vs T3 and T3 vs T4); vertical black bars: 95% confidence intervals; T2: the parturition day; T3: twenty-one days after parturition; T4: two months after parturition.

lower hair cortisol concentrations at T4 compared to dams who maintained or gained weight; $P$ = 0.037). Parity, pregnancy and delivery duration, as well as colostrum/milk composition were not linked to maternal nor neonatal cortisol concentrations. In pups, body weight, litter size and gender were not related to cortisol concentrations. However, healthy pups (n = 20) showed lower salivary and hair cortisol concentration than pups with congenital defects (n = 2), even though these data were not statistically relevant due to the small number of pathological pups. Specifically, at birth (T2) healthy pups showed median cortisol concentrations of 0.4 μg/dL in saliva and of 0.008 μg/dL in hair, compared to 1.43 μg/dL and 0.13 μg/dL respectively in saliva and hair, of the two malformed pups.

## Discussion

The literature on cortisol measurement in both adult dogs and pups is affected by the heterogeneity of analytical methods making difficult any comparative quantitative review [22, 27–29]. Large discrepancies in cortisol ranges and measurement units have been published [22, 28–31], especially for hair, probably due to the different methods used and different hair pigmentation. For this reason, in this study hair from the ventral surface of the tail base was collected, since in German shepherd dogs light hair growths in this region. Maternal hair was shaved three times at about 2 months interval, as shorter collection time resulted in an insufficient amount of re-growth hair. Although the complete hair growth cycle takes 6 to 12 weeks depending on the breed [22], the rate of hair growth was faster in pups, allowing three sampling in a shorter time period.

Plasma [32, 33] and salivary [22, 34–36] cortisol concentrations measured in dams agreed with the values reported in healthy adult dogs. In our caseload, the highest cortisol values were detected in maternal plasma. However, the difference between plasma and salivary cortisol levels in dams was significant only at parturition. In people, the strong correlation between salivary and plasma cortisol concentration makes salivary cortisol the best marker of adrenocortical function [27]. As expected, the highest plasma and salivary (but not hair) cortisol values were measured in one bitch involved in a minor car collision a couple of hours before the sampling (ID. 4). Indeed, both blood and salivary cortisol concentrations reflect the punctual adrenocortical response to acute stress, while chronic stress is mostly detected by hair which acts as a cortisol storage site [16, 22].

In people is reported that cortisol concentrations can be affected by circadian fluctuations. Specifically, in non-stressful conditions, healthy adult humans show the highest plasma cortisol concentrations in the morning that gradually decrease till midnight [1, 16]. For this reason, even though there is a lack of consensus about circadian cortisol fluctuation in dogs, the sampling in the present study was always performed between 8:00 and 12:00 AM, in order to reduce possible confounding factors. A stress-induced increase in cortisol levels is not expected to occur before 30 minutes from the stressful event [34, 37]. Therefore, biological samplings were collected with minimal restraint and completed within five minutes, which was enough to prevent elevation in cortisol levels caused by stress [1, 22].

The present study demonstrates that cortisol measurement can be easily performed in dog saliva and hair in both dams and pups, whereas the low cortisol content in canine milk and colostrum suggests the need of a more sensitive assay method for this matrix. The small saliva and hair amounts needed to evaluate cortisol concentrations, together with the non-invasive collection technique, made these matrices a feasible alternative to whole blood, decreasing the discomfort that may compromise the interpretation of results, especially in neonate and uncooperative patients [8]. Similarly, milk volume required for cortisol measurement was negligible compared to milk production in dogs [38, 39] and its collection caused a minimal disturbance

to the animals. However, some milk samples of our caseload showed cortisol concentration below the detection limit. This could depend on the milk fat percentage which may interfere with the ELISA analysis if the sample is not skimmed adequately. The skimming and cortisol extraction procedures used in this study were adapted from researches performed on other species [23] since no reports were available in dogs. Milk cortisol values obtained in dogs were lower than those reported in women, goats and cattle [2, 8, 40]. Interspecific differences in diet and environment could affect milk composition and in turn cortisol level or detection [40]. Unfortunately, the small number of milk samples that showed a cortisol concentration above the detection limit threshold do not allow to perform a statistical comparison. However, the feasibility of milk sampling together with its influence on the behavioural phenotype development, infant's innate and adaptative immune system, energy uptake and weight gain as shown in humans and primates [2, 41–43], and on productive performances as was observed in dairy cows [18], deserve further insights in canine species.

The assessment of salivary cortisol concentration is a precise method to indicate neonatal stress in humans [44]. Babies exposure to stress has been associated with increased cortisol concentrations which may result in a greater risk of mortality [45]. So far, salivary and hair cortisol reference values have not been described in live neonatal and paediatric pups. In the present study, salivary cortisol concentration was significantly higher at birth (T2) and after 2 months (T4) than at 21 days of life (T3) in pups. This result is similar to what observed in human neonates, where serum cortisol concentrations are higher at the time of birth, due to the stress resulting from labour, as a necessary neonatal adaptative system to extra-uterine life and then decrease due to the rapid involution of adrenal glands cortisol levels [46]. The subsequent further increase in neonatal salivary cortisol recorded at T4 could be associated with the weaning period. Indeed, during this period, several stressors such as dietary change to semi-solid food, littermates competition and separation from the mother occur. Interestingly, in two pups with malformations cortisol concentration in both saliva and hair was higher compared to healthy pups. Despite the small number of pathological pups in our caseload did not allow statistical evaluation, this cortisol increase may reflect neonatal suffering even before the presentation of overt clinical signs. Thus, cortisol concentration may be a potential early marker for detection of subclinical pathological conditions in neonatal canine patients. Further studies on a larger caseload are required to confirm this hypothesis.

Preliminary results obtained in this study failed to find correlation between cortisol concentrations and some clinical features. For example, aging is expected to alter the function of the hypothalamic-pituitary-adrenal axis, as reported in people [47]. Nevertheless, in the present study changes in maternal cortisol concentration depending on age were not observed. A possible explanation is the small sample size of our caseload. Despite a cortisol increase due to prenatal maternal stress is related with prematurity and low birthweight in babies [4, 7], no correlation between maternal cortisol values and neonatal body weight was found in this study. It should be noted that the dams enrolled lived in a quiet environment throughout the survey period and all pups were born at term by vaginal delivery without signs of prematurity. Therefore, this lack of correlation was consistent with environmental conditions of dams and it is also underlined by the lack of differences in cortisol measurement among collection times in any matrix. Some authors suggested that cortisol secretion may be oestradiol-dependent thus, gender dependent [48]. However, gender did not influence cortisol concentration in saliva and hair of pups in the present study, probably due to their immature prepuberal hormonal status. Similarly, no significant correlation was found between cortisol concentration and BCS, even though dams that lost weight after weaning had lower hair cortisol concentration compared to the others bitches. In obese people higher levels of cortisol are reported [48]. Although none of the dams enrolled in this study was obese, it is possible that body weight and

BCS can affect cortisol levels in dogs as well and this hypothesis is worthy to be better investigated on a larger caseload. The results obtained in this study could represent a starting point for the diagnostic use of this hormone in dogs by providing physiological values and procedures to measure cortisol in innovative and feasible matrices.

## Conclusions

Cortisol measurement may help in early recognition of stress and monitoring neonatal pups, providing clinician new tools to take care of fragile patients. A possible use of this biomarker could apply to quantify effects of transport condition in the pup trade or to improve canine breeding management. The use of a polyparous species such as the dog allows speculations on the clinical significance of this potential marker, reducing the bias risk intrinsic to primates and humans. The small sample size of the present study is a limitation that cannot be neglected in generalizing these results. Thus, future studies on a larger caseload are recommended to confirm the results here obtained. Moreover, the impact of stressful environment, pathological pregnancy and dystocic parturition, together with neonatal suffering could be investigated.

## Supporting information

**S1 Table. Cortisol concentration raw data.** In the sheet named "Cortisol" are presented all the raw data for cortisol concentration (μg/dL) at each time point and matrix, for both dams and pups. Specifically, in the column ID, each ID. N corresponds to a dam, and each N.1, N.2. . . corresponds to its pups.
(XLSX)

**S2 Table. Milk composition raw data.** In the sheet named "Milk composition" are presented all the raw data concerning milk energy and composition in terms of protein, fat and carbohydrates for each colostrum/milk sample collected.
(XLSX)

## Author Contributions

**Conceptualization:** Debora Groppetti.

**Data curation:** Alessandro Pecile.

**Formal analysis:** Sara Meazzi, Sergio A. Zanzani.

**Funding acquisition:** Debora Groppetti, Alessandro Pecile.

**Investigation:** Debora Groppetti, Sara Meazzi, Joel F. S. Filipe, Carla Colombani, Sara Panseri, Clara Palestrini, Simona Cannas, Alessandro Pecile.

**Methodology:** Clara Palestrini, Alessia Giordano.

**Supervision:** Alessandro Pecile.

**Validation:** Joel F. S. Filipe, Carla Colombani, Sara Panseri.

**Visualization:** Sergio A. Zanzani.

**Writing – original draft:** Debora Groppetti, Sara Meazzi, Alessia Giordano.

**Writing – review & editing:** Debora Groppetti, Sara Meazzi, Alessia Giordano.

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
