## [Decision Letter · Decision Letter 0]

21 Apr 2021

PONE-D-21-05956

Maternal and neonatal canine cortisol measurement in multiple matrices during the perinatal period: a pilot study

PLOS ONE

Dear Dr. Giordano,

Thank you for submitting your manuscript to PLOS ONE. After careful consideration, we feel that it has merit but does not fully meet PLOS ONE’s publication criteria as it currently stands. Therefore, we invite you to submit a revised version of the manuscript that addresses the points raised during the review process.

We look forward to receiving your revised manuscript.

Kind regards,

Benito Soto-Blanco, DVM, MSc, PhD

Academic Editor

PLOS ONE

2. In your Methods section, please include a comment about the state of all the animals following this research.

Reviewers' comments:

Reviewer's Responses to Questions

**Comments to the Author**

1. Is the manuscript technically sound, and do the data support the conclusions?

Reviewer #1: Partly

Reviewer #2: Yes

Reviewer #3: Yes

2. Has the statistical analysis been performed appropriately and rigorously? 

Reviewer #1: No

Reviewer #2: Yes

Reviewer #3: Yes

3. Have the authors made all data underlying the findings in their manuscript fully available?

Reviewer #1: No

Reviewer #2: Yes

Reviewer #3: Yes

4. Is the manuscript presented in an intelligible fashion and written in standard English?

Reviewer #1: Yes

Reviewer #2: No

Reviewer #3: Yes

5. Review Comments to the Author

Reviewer #1: After reading the manuscript carefully, I guess that the work has an interesting assumption and is important from a practical point of view, but unfortunately it is burdened with a number of methodological and analytical shortcomings and errors. I suggest you carefully analyze and change the methodology and correctly select the analytical methods for the biological material. In addition, Authors should to create a hypothesis and correctly describe the results, in accordance with the canons adopted in scientific works.

Below I present content-related and technical comments to the manuscript:

The title is not very precise and corresponding to the contents of the manuscript and there key words are not adequate to manuscript.

Introduction

Line 55-60: The work lacks a research hypothesis and a clear purpose of the work. The reader must guess for himself what the Authors wanted to investigate / evaluate. I believe that a scientific work should clearly formulate a hypothesis and then a clear, concise research goal. Such a presentation also greatly facilitates the understanding of the Authors' way of thinking. In addition, the purpose of the work is unclear and to poorly defined.

Line 65-66: In my opinion, 5 individuals are not enough to be considered representative of this group in this study. Moreover, since we do not analyze the influence of breed on the assessed indicators (here cortisol), it was perfectly possible to use dogs of other breeds to increase the size of the research group. Even the pilot studies referred to by the authors require much greater involvement, because we will then base them methodically on them.

Line 66-68: It seems to me that the presence of animals in a different environment than the home where the dog stays on a daily basis is an additional stress factor and therefore such factor disqualifies this group (and so small, because only 5 individuals) as a representative research group.

Table 1: For graphic improvement. Row 1 should begin equally in all four columns.

Line 80-81: Please describe the saliva collection tool in detail and describe the method of collection. The current form of the message is too laconic. In addition, it is not stated how long the saliva was collected, who was present at the time of saliva collection (these are also important stress factors for the animal).

Line 91-94: The methodology lacks basic information on how the hair sample was prepared for further analysis, especially since the Elisa test was designed for saliva testing. By default, Elisa tests do not allow the use of hair as a research material. The mention in the next paragraph to work no. 20 is an overly disrespectful treatment of this topic. The reviewer is familiar with the quoted publication No. 20 and has many doubts as to the correctness of the modification proposed there for the Elisa test. In addition, there is no exact information whether cortisol or its metabolite was determined in the hair; in many cases we generally say that a given analyte is analyzed when in fact we determine its metabolite, e.g. formed during biotransformation. Moreover, the question arises which patterns were used? There is also no information on the adaptation of the Elis method to saliva cortisol analysis for hair samples. In my opinion, improper analytical procedure was used, it is unjustified and, unfortunately, due to the maladjustment of the biological material to the conditions of the method (provided by the manufacturer of the test), it may be subject to a large error. The authors also do not state whether the test used was directed at dogs. The manufacturer of the test states that the test is intended for humans and "certain animals", but does not specify for which ones. Therefore, we do not know if it is dedicated to dogs, rats or for poultry. Therefore, on what basis did the authors choose this test, not being sure that there is a species compatibility? This is another very big analytical error, because the composition of the test reagents is strictly adapted to the species, except for poultry.

There is also no information on the amount of sample of biological material used in the Elisa test. Generally, cortisol levels are given in the unit nmol / L or ng / mL, not µg / dL. The order of magnitude of the cortisol level is also of concern.

Line 102-106: What was the purpose of analyzing the composition of the milk?

Line 107-121: I propose to use other statistical models. According to the reviewer, it would be more reasonable to use two-factor statistics comparing specific features, e.g. the influence of maternal age and the influence of time, on cortisol levels. This is just an example. And, unfortunately, there is a problem with the size of the group, which in my opinion is definitely too small for statistical anlysis. However, it is reasonable to assess the correlation proposed by the authors.

Line 124-135: Results are incorrectly described. Figures are placed in tables so as not to be quoted in the Results section! I do not understand why to compare the level of cortisol in different material (no hypothesis), since it is not reflected in practice.

Table 2: It is not possible to read in Table 2 what statistically significant differences were obtained. On what basis, then, the authors write about them in the text? Tables are unreadable.

Discussion

In the discussion, there is no reference to the results of other researchers, and there is too much general knowledge.

Conclusion

It is difficult to talk about the conclusions of in this manuscript due to the lack of a hypothesis and a proper purpose of the work. The presented conclusions are an attempt to gather information on this subject, but they are certainly not conclusions from the work. Conclusions should clearly convey information and give advice / recommendation to other researchers / practitioners for the future.

Reviewer #2: The study described in this manuscript appears sound and well thought through. Additionally I am positive that the results presented in this manuscript can contribute to the understanding of neonatal stress and welfare in dogs. I would therefore like to thank the authors for their contribution to this field of science.

The manner in which the manuscript is written however can be improved. Regarding the contents of the manuscript, it would be informative for readers to have more context or background information at certain points (see also specific issues for further clarification). Regarding the use of the English language itself, there are frequent instances of grammatical errors, and sentences which are not fully logical in construction (not further specified below due to their number). I would therefore highly recommend to have this manuscript evaluated by an English language expert or native speaker.

Specific comments:

Line 66: I was wondering by what standards dogs where judged for health. Was this done by a veterinarian or other professional?

Line 67: Is it possible to give more information on the environmental conditions in which the dogs were housed and if the dogs were habituated to these environments prior to the study? Various factors in a dog's environment are known to influence their cortisol level? Was housing indoor or outdoor or a mix of the two? Were the dogs exercised and how often? Was enrichment available? And finally, was contact with con-specifics or humans possible?

Line 101: Is it possible to give intra- (and inter- if applicable) assay coefficients for the cortisol immuno assay kits used during this study?

Line 168: It would be informative to know what the exact p-value was even though it was not significant. The level of non-significance might be informative for future research (was a trend observable or was non-significance high).

Line 175: see comment to line 168.

Line 179: see comment to line 168.

Line 195: Is the restraint that is needed to draw blood of an uncooperative animal truly heavier than the restrained that is needed for a saliva sample or hair sample from that same animal?

Line 199: how do the fat and protein percentages of dog milk compare to those of other mammals? Were these differences accounted for in sample preparation during this study?

Line 247: Though a circadian rhythm in cortisol is indeed found in humans and most other mammals, such a rhythm has not been found in dogs during several studies. Did the authors take this into consideration when they chose their time frame for measurements and do they think it might have affected measurements.

Line 254: I might have overlooked this point in the result section so please ignore the following statement if so: Did the authors also consider the status nulliparous or parous besides age as a possible stress influencing factor surrounding birth? It was mentioned that this was not the first litter for all dogs so this factor might have influenced results.

Reviewer #3: That is a really interesting and innovative study that presents new matrices for cortisol evaluation in canine dams and puppies. The contents are excelent, but text lacks a better organization and it needs to be rewritten in some cases. At my point of view, manuscript deserves to be published, but only after a careful revision based on the points suggested below.

Tittle - It is adequate

Abstract - It is well written and summarizes the methods and the main findings of the research.

Introduction - Even if this topic present some valuable information, it lacks a better organization. At the begining of the Introduction, a general paragraph presenting those general information related to mammals is really welcome. But after this, authors should present, in a new paragraph, what is known about this subject in dogs by now, what is missing? why this knowledghe would be so important for the species? How knowing other matrices for cortisol measurements would be interesting? Then, a last paragraph presenting your innovative proposal, would be welcome. Please, organize your ideas in different paragraphs for a better reader comprehension.

Furthermore, at the end of your objectives, authors says: some clinical aspects in German shepherd dogs were investigated. Why is this important? Please, provide some considerations in your introduction to justify the inclusion of this aim.

Material and Methods -

- Please review the guidelines for presenting Tables. They should not contain internal boards.

- At T2, the day of the parturition, shouldn't be milk called as collostrum? These are completely different substances in terms or composition. Please revise this in all the text.

- The text for Biological Samples Collections seems a few confuse. Please, separate in different paragraphs the methods for collection of each kind of sample (Milk and colostrum, blood, hair, etc)

- Regarding cortisol assay, were the intra- and inter-assay variability evaluated? Provide the values.

- What is the aim for analysing milk composition? There is no information at the introduction that support the importance of this analysis at the present study.

Results

- The values reported at the first paragraph, related to some aspects of the mothers and the parturition, should be better organized in a table presenting mean values and ranging.

- Lines 136 to 154 related to Cortisol Measure in dams is sometimes difficult to comprehend. Please, reorganize your ideas. Try to present the results in a more understandble form to the reader. The Table 2 is incomprehensible. What is interquatile? Why n = 67 for dams and n=128 for puppies? Why put all samples together?

- Why milk composition appears at the topic related to Cortisol Measurement in Dogs? What is the importance of this information for the present study?

- Regarding correlation of cortisol with clinical features, a presentation in Table or Graph would be welcome. Text is not clear.

Discussion

- Even if discussion present relevant information and it clearly discuss the results. It lacks a better organization. Please, separe different ideas in different paragraphs.

- The explanation for analysing milk composition only appears at the discussion (lines 203 - 205). It should be included at the introduction.

Conclusions are clearly supported by the results.

References are adequate.

6. PLOS authors have the option to publish the peer review history of their article (what does this mean?). If published, this will include your full peer review and any attached files.

Reviewer #1: No

Reviewer #2: **Yes: **Emmy A.E. van Houtert MSc

Reviewer #3: No

---

## [Author Response · Author response to Decision Letter 0]

10 Jun 2021

PONE-D-21-05956

Maternal and neonatal canine cortisol measurement in multiple matrices during the perinatal period: a pilot study

PLOS ONE

5. Review Comments to the Author

Reviewer #1: After reading the manuscript carefully, I guess that the work has an interesting assumption and is important from a practical point of view, but unfortunately it is burdened with a number of methodological and analytical shortcomings and errors. I suggest you carefully analyze and change the methodology and correctly select the analytical methods for the biological material. In addition, Authors should to create a hypothesis and correctly describe the results, in accordance with the canons adopted in scientific works.

AA: We would like to thank the Reviewer for his/her comments that help us to improve the quality of this manuscript. Some interesting points were raised, which make the text clearer and easier to understand.

Below I present content-related and technical comments to the manuscript:

The title is not very precise and corresponding to the contents of the manuscript and there key words are not adequate to manuscript.

AA: The keywords have been changed as suggested by the reviewer. Since the other two reviewers did not arise any issue concerning the title, we would like to maintain the current one, because it seems appropriate to the content of the manuscript, unless this reviewer would like to further detailed his/her suggestion. 

Introduction

Line 55-60: The work lacks a research hypothesis and a clear purpose of the work. The reader must guess for himself what the Authors wanted to investigate / evaluate. I believe that a scientific work should clearly formulate a hypothesis and then a clear, concise research goal. Such a presentation also greatly facilitates the understanding of the Authors' way of thinking. In addition, the purpose of the work is unclear and to poorly defined.

AA: Thanks for your suggestion. Scientific hypotheses have been added in the introduction. 

Line 65-66: In my opinion, 5 individuals are not enough to be considered representative of this group in this study. Moreover, since we do not analyze the influence of breed on the assessed indicators (here cortisol), it was perfectly possible to use dogs of other breeds to increase the size of the research group. Even the pilot studies referred to by the authors require much greater involvement, because we will then base them methodically on them.

AA: We agree with the reviewer that the caseload size is quite small (and this point was addressed as a limitation of the study in the Discussion section). However, based on statistical studies (doi: 10.1177/2192568218782679), the minimum and sufficient number to have analyzable data is five subject. Moreover, samples from the five dams and theirs 22 pups were collected at five and three time-point respectively. Thus, cortisol was measured on a total of 75 maternal samples and 207 pup samples. We think that these results can represent a useful starting point for future large-scale studies. Moreover, in human studies, mother-child cortisol comparisons are made on a single born (as monotocyc species), while dogs allow to avoid these bias due to the deliver of more pups per litter.

The choice of a single canine breed has the aim to reduce the factors that may interfere with cortisol results, such as genetic influences temperament and the stress tolerance level. This consideration has been added in the material and methods section. 

Line 66-68: It seems to me that the presence of animals in a different environment than the home where the dog stays on a daily basis is an additional stress factor and therefore such factor disqualifies this group (and so small, because only 5 individuals) as a representative research group. 

AA: Actually, the dogs lived together in the same breeding facility before, during and after this study. For this reason, we do not think that this could be a stressor for them. This point has been better clarified in the manuscript. 

Table 1: For graphic improvement. Row 1 should begin equally in all four columns.

AA: Thank you for the suggestion. We take the chance to modify all the tables by removing the vertical lines. Hope this could be fine. 

Line 80-81: Please describe the saliva collection tool in detail and describe the method of collection. The current form of the message is too laconic. In addition, it is not stated how long the saliva was collected, who was present at the time of saliva collection (these are also important stress factors for the animal).

AA: Thanks for your comment. We better specified in the manuscript the method of saliva collection. 

Line 91-94: The methodology lacks basic information on how the hair sample was prepared for further analysis, especially since the Elisa test was designed for saliva testing. By default, Elisa tests do not allow the use of hair as a research material. The mention in the next paragraph to work no. 20 is an overly disrespectful treatment of this topic. The reviewer is familiar with the quoted publication No. 20 and has many doubts as to the correctness of the modification proposed there for the Elisa test. In addition, there is no exact information whether cortisol or its metabolite was determined in the hair; in many cases we generally say that a given analyte is analyzed when in fact we determine its metabolite, e.g. formed during biotransformation. Moreover, the question arises which patterns were used? There is also no information on the adaptation of the Elis method to saliva cortisol analysis for hair samples. In my opinion, improper analytical procedure was used, it is unjustified and, unfortunately, due to the maladjustment of the biological material to the conditions of the method (provided by the manufacturer of the test), it may be subject to a large error. The authors also do not state whether the test used was directed at dogs. The manufacturer of the test states that the test is intended for humans and "certain animals", but does not specify for which ones. Therefore, we do not know if it is dedicated to dogs, rats or for poultry. Therefore, on what basis did the authors choose this test, not being sure that there is a species compatibility? This is another very big analytical error, because the composition of the test reagents is strictly adapted to the species, except for poultry.

There is also no information on the amount of sample of biological material used in the Elisa test. Generally, cortisol levels are given in the unit nmol / L or ng / mL, not µg / dL. The order of magnitude of the cortisol level is also of concern.

AA: The Reviewer is right, the article of Sauvè et al. (former reference 20), describes the use of an ELISA kit for salivary cortisol to measure hair cortisol, but it is not the same kit used in the present study. Nevertheless, the use of this specific ELISA kit for measurement of canine cortisol on hair, other than saliva, was already reported in literature (including Bennett et Hayssen, 2010). For what concern the unit used, the same article (now Reference 22) use pg/dL for hair cortisol and microg/dL for saliva cortisol. The use of the same unit for all the cortisol measurement in the present study was chosen in order to allow a comparison among the different substrates. The reference of Sauvè et al. has been replaced since, as the Reviewer suggested, although focused on measurement of hair cortisol using a salivary ELISA kit, did not use the same ELISA kit. 

As reported in material and methods section, the amount of hair shaved by each pup was a minimum of 8 mg, in order to not cause discomfort. The protocol used was similar to those of Sauvè et al and Bennett and Hayssen (of course a proportion based on the amount of collected hair was applied). The second reference has been added as well.

Line 102-106: What was the purpose of analyzing the composition of the milk?

AA: In human, milk composition has been associated with cortisol concentration. “Mainly in behavioral studies, it is recommended a simultaneous evaluation of milk energy and yield together with cortisol titration (Hinde et al., 2015)”. This information has been added in the text.

Line 107-121: I propose to use other statistical models. According to the reviewer, it would be more reasonable to use two-factor statistics comparing specific features, e.g. the influence of maternal age and the influence of time, on cortisol levels. This is just an example. And, unfortunately, there is a problem with the size of the group, which in my opinion is definitely too small for statistical anlysis. However, it is reasonable to assess the correlation proposed by the authors.

AA: The statistical approach chosen was the one proposed by our statistician, because it was the most suitable for the comparison performed. Of course, the sample size is small (as we reported also as a limitation in the discussion section). Anyway, even if limited to few results, statistical significance was found and could be further confirmed in future studies on a larger caseload.

Line 124-135: Results are incorrectly described. Figures are placed in tables so as not to be quoted in the Results section! I do not understand why to compare the level of cortisol in different material (no hypothesis), since it is not reflected in practice.

AA: The comparison among different matrices was performed in order to evaluate the magnitude of cortisol measurement. Of course, the biological meaning of salivary and hair cortisol is different and reflect acute and chronic stress respectively. This point has been better specified both in the introduction and discussion section.

Figure caption was formatted accordingly to PloSONE guidelines, while the figure files could be found as a separate file (the Reviewer could find it at the end of the manuscript pdf file): “Figure captions are inserted immediately after the first paragraph in which the figure is cited. Figure files are uploaded separately. Tables are inserted immediately after the first paragraph in which they are cited.”

Table 2: It is not possible to read in Table 2 what statistically significant differences were obtained. On what basis, then, the authors write about them in the text? Tables are unreadable.

AA: Table 2 is referred to the mean cortisol values obtained for each substrate (as reported in the title of the Table). Statistically significant results among different matrices was reported in Figure 1 (that could be found at the end of the pdf file). 

Discussion

In the discussion, there is no reference to the results of other researchers, and there is too much general knowledge.

AA: Discussion section has been reorganized so now results are discussed in the same order in which they are presented in the Result section. We take the chance to do some slight modification to English language and to reduce some general knowledge. We hope that now this section is easier to understand. 

Conclusion

It is difficult to talk about the conclusions of in this manuscript due to the lack of a hypothesis and a proper purpose of the work. The presented conclusions are an attempt to gather information on this subject, but they are certainly not conclusions from the work. Conclusions should clearly convey information and give advice / recommendation to other researchers / practitioners for the future.

AA: The hypothesis has been better expressed in the Introduction section. Conclusion have been summarized accordingly. 

Reviewer #2: The study described in this manuscript appears sound and well thought through. Additionally I am positive that the results presented in this manuscript can contribute to the understanding of neonatal stress and welfare in dogs. I would therefore like to thank the authors for their contribution to this field of science.

The manner in which the manuscript is written however can be improved. Regarding the contents of the manuscript, it would be informative for readers to have more context or background information at certain points (see also specific issues for further clarification). Regarding the use of the English language itself, there are frequent instances of grammatical errors, and sentences which are not fully logical in construction (not further specified below due to their number). I would therefore highly recommend to have this manuscript evaluated by an English language expert or native speaker.

Specific comments:

Line 66: I was wondering by what standards dogs where judged for health. Was this done by a veterinarian or other professional? 

AA: Bitches enrolled in this study are high genealogy selected German shepherd dogs intended for breeding. All the dogs underwent routine examination before breeding, that includes the evaluation of regular vaccinations and prophylaxis, CBC and biochemistry test, DNA deposit, X-Ray to exclude HD and ED, and accurate reproductive cycle monitoring. All the dogs were tested for working and aptitude performance according to Italian Schäferhunde Society (SAS) regulations. All the dams had been monitored by a veterinarian specialized in reproduction during the whole pregnancy duration. The same veterinarian performed a clinical evaluation of all the pups during the study period. Information about the health evaluation for each dog was briefly added in the text. Thank you for the comment. 

Line 67: Is it possible to give more information on the environmental conditions in which the dogs were housed and if the dogs were habituated to these environments prior to the study? Various factors in a dog's environment are known to influence their cortisol level? Was housing indoor or outdoor or a mix of the two? Were the dogs exercised and how often? Was enrichment available? And finally, was contact with con-specifics or humans possible?

AA: Sorry for being unclear. We meant that dogs enrolled in this study lived in the same breeding facility for the entire study period. All the samplings were performed at the breeding facility. This point was better clarified in the manuscript. 

Line 101: Is it possible to give intra- (and inter- if applicable) assay coefficients for the cortisol immuno assay kits used during this study? 

AA: The information requested were added in the manuscript.

Line 168: It would be informative to know what the exact p-value was even though it was not significant. The level of non-significance might be informative for future research (was a trend observable or was non-significance high).

AA: The P-value for the comparison between T2 and T4 in puppies has been added in the manuscript.

Line 175: see comment to line 168.

Line 179: see comment to line 168.

AA: In those specific cases the total number of P-value was really high and it would be impossible to add all this information in the text. We perform an evaluation for each time point and each matrix. The only possible way would be to add a Table which is totally non informative because all the P-value are above 0.1. Nevertheless, if the reviewer and the editor feel that those information are needed, we could add it as Supporting information. 

Line 195: Is the restraint that is needed to draw blood of an uncooperative animal truly heavier than the restrained that is needed for a saliva sample or hair sample from that same animal?

AA: Hair and saliva are alternative substrates, mainly intended for uncooperative animals (human included). As an example, in Zoological Parks captive wild animals are trained to open the mouth or to be gently touched by keepers. Of course, this is not enough to perform a blood sampling without sedation/anesthesia but it is sufficient for a saliva or hair sampling. Moreover, both hair and saliva samplings are not painful, thus better tolerated by dogs. 

Line 199: how do the fat and protein percentages of dog milk compare to those of other mammals? Were these differences accounted for in sample preparation during this study?

AA: Sorry, we are not sure to understand your question. We did not compare milk composition between dogs and other mammals as it is well known milk composition is species-specific. The analytic methods to measure chemicals do not vary by species but by the nature of the substance. In particular, the method to measure lipids in milk was described by Collares et al., 1997 and applied to different species. The sample total protein count was determined by using a spectrophotometric method with micro kit (Sigma) and spectrophotometer (VWR UV 31000 PC). The procedure, a dye micro technique (Brilliant Blue G, Sigma), is based on method described by Bradford et al., 1976 and applied to different species as well. Specifically, the reference for cortisol extraction from skimmed milk was referred to cattle. Fat milk percentage in cows is about 3-4%, whereas in dogs is reported to be over 9% (Lactation in the dog: milk composition and intake by puppies, Oftedal, 1984). For this reason, we hypothesized that fat content may interfere with ELISA cortisol measurement. Discussion has been re-organized and partially re-written, we hope it is more clear now.

Line 247: Though a circadian rhythm in cortisol is indeed found in humans and most other mammals, such a rhythm has not been found in dogs during several studies. Did the authors take this into consideration when they chose their time frame for measurements and do they think it might have affected measurements.

AA: We agree with the Reviewer, there is no consensus regarding the circadian rhythm of cortisol secretion in dogs. However, some authors reported higher cortisol concentrations in the morning (Kolevská and Brunclín, 2002). We decided to perform sampling at the same time range in order to reduce a possible source of variability. 

Line 254: I might have overlooked this point in the result section so please ignore the following statement if so: Did the authors also consider the status nulliparous or parous besides age as a possible stress influencing factor surrounding birth? It was mentioned that this was not the first litter for all dogs so this factor might have influenced results.

AA: Yes, we recorded this data. Two bitches were nulliparous and the remaining three dogs did one to four previous parturitions (former LL 127-128). Unfortunately, due to the limited size of our caseload a statistical comparison was not feasible. 

Reviewer #3: That is a really interesting and innovative study that presents new matrices for cortisol evaluation in canine dams and puppies. The contents are excelent, but text lacks a better organization and it needs to be rewritten in some cases. At my point of view, manuscript deserves to be published, but only after a careful revision based on the points suggested below.

Tittle - It is adequate

Abstract - It is well written and summarizes the methods and the main findings of the research.

Introduction - Even if this topic present some valuable information, it lacks a better organization. At the begining of the Introduction, a general paragraph presenting those general information related to mammals is really welcome. But after this, authors should present, in a new paragraph, what is known about this subject in dogs by now, what is missing? why this knowledghe would be so important for the species? How knowing other matrices for cortisol measurements would be interesting? Then, a last paragraph presenting your innovative proposal, would be welcome. Please, organize your ideas in different paragraphs for a better reader comprehension.

Furthermore, at the end of your objectives, authors says: some clinical aspects in German shepherd dogs were investigated. Why is this important? Please, provide some considerations in your introduction to justify the inclusion of this aim.

AA: Introduction has been edited based on the Reviewer’s suggestions. We hope that now it is better organized and readable.

Material and Methods -

- Please review the guidelines for presenting Tables. They should not contain internal boards.

AA: Thank you for noticing. We modified the tables accordingly. 

- At T2, the day of the parturition, shouldn't be milk called as collostrum? These are completely different substances in terms or composition. Please revise this in all the text.

AA: It was mentioned only in the first paragraph about biological samples collection. Thank you for your suggestion. We renamed it also in Table1 and in the “Cortisol assay” paragraph.

- The text for Biological Samples Collections seems a few confuse. Please, separate in different paragraphs the methods for collection of each kind of sample (Milk and colostrum, blood, hair, etc)

AA: Thank you for your suggestion, this section has been reorganized according to your comment.

- Regarding cortisol assay, were the intra- and inter-assay variability evaluated? Provide the values.

AA: The values have been now reported in the manuscript.

- What is the aim for analysing milk composition? There is no information at the introduction that support the importance of this analysis at the present study.

AA: Thank you for your comment. We add some information in the Introduction section and we hope that now it is clearer.

Results

- The values reported at the first paragraph, related to some aspects of the mothers and the parturition, should be better organized in a table presenting mean values and ranging.

AA: Thank you for your comment. Table 2, containing most of the information about signalment and parturition for each dams has now been added. 

- Lines 136 to 154 related to Cortisol Measure in dams is sometimes difficult to comprehend. Please, reorganize your ideas. Try to present the results in a more understandble form to the reader. The Table 2 is incomprehensible. What is interquatile? Why n = 67 for dams and n=128 for puppies? Why put all samples together?

AA: The comparison among different matrices was performed in order to evaluate the magnitude of cortisol concentration. For the same reason each sampling of the same matrix was considered as independent. 

n = 67 and n = 128 for dams and puppies respectively is referred to the total number of samples analysed. However, the Reviewer is right, this number could confound the reader. For this reason, we modify the table and add the number of total samples for each matrix. Due to the low number of sampling, the distribution of data is not parametric, thus statistic software suggest that the better expression of results would be as median and interquartile range. 

- Why milk composition appears at the topic related to Cortisol Measurement in Dogs? What is the importance of this information for the present study?

AA: Some information about this point has been added in the Introduction section. 

- Regarding correlation of cortisol with clinical features, a presentation in Table or Graph would be welcome. Text is not clear.

AA: Thank you for your suggestion. The paragraph has been modified and hopefully is now clearer. Since the data and statistics are really a huge amount of numbers, but none of it is statically relevant, if the reviewer and the editor agree, we could add these data as supporting information. 

Discussion

- Even if discussion present relevant information and it clearly discuss the results. It lacks a better organization. Please, separe different ideas in different paragraphs.

- The explanation for analysing milk composition only appears at the discussion (lines 203 - 205). It should be included at the introduction. 

AA: Thank you for your comment and suggestions. Discussion section has been reorganized so now results are discussed in the same order in which they are presented in the Result section. We take the chance to do some slight modification to English language, we hope that now this section is easier to understand.

Conclusions are clearly supported by the results.

References are adequate.

---

## [Decision Letter · Decision Letter 1]

5 Jul 2021

Maternal and neonatal canine cortisol measurement in multiple matrices during the perinatal period: a pilot study

PONE-D-21-05956R1

Dear Dr. Giordano,

We’re pleased to inform you that your manuscript has been judged scientifically suitable for publication and will be formally accepted for publication once it meets all outstanding technical requirements.

Kind regards,

Benito Soto-Blanco, DVM, MSc, PhD

Academic Editor

PLOS ONE

Additional Editor Comments (optional):

Reviewers' comments:

Reviewer's Responses to Questions

**Comments to the Author**

1. If the authors have adequately addressed your comments raised in a previous round of review and you feel that this manuscript is now acceptable for publication, you may indicate that here to bypass the “Comments to the Author” section, enter your conflict of interest statement in the “Confidential to Editor” section, and submit your "Accept" recommendation.

Reviewer #1: (No Response)

Reviewer #2: All comments have been addressed

Reviewer #3: All comments have been addressed

2. Is the manuscript technically sound, and do the data support the conclusions?

Reviewer #1: Partly

Reviewer #2: Yes

Reviewer #3: Yes

3. Has the statistical analysis been performed appropriately and rigorously? 

Reviewer #1: Yes

Reviewer #2: Yes

Reviewer #3: Yes

4. Have the authors made all data underlying the findings in their manuscript fully available?

Reviewer #1: Yes

Reviewer #2: Yes

Reviewer #3: Yes

5. Is the manuscript presented in an intelligible fashion and written in standard English?

Reviewer #1: Yes

Reviewer #2: Yes

Reviewer #3: Yes

6. Review Comments to the Author

Reviewer #2: Overall:

I would like to app0laud the authors in their efforts to incorporate all reviewer comments. I can clearly see alterations at various points in the manuscript and therefore think that comments were adequately addressed.

Grammar and spelling:

Although the writing still contains grammatical errors and is inconsistent at certain points i do believe that the text is readable and understandable at this point. I would therefore leave it up to the editor whether or not another English language revision is required.

Reviewer #3: Authors addressed to all my previous suggestions. At my point of view, manuscript can now be accepted for publication.

7. PLOS authors have the option to publish the peer review history of their article (what does this mean?). If published, this will include your full peer review and any attached files.

Reviewer #1: **Yes: **MAGDALENA KRAUZE

Reviewer #2: **Yes: **Emmy van Houtert

Reviewer #3: No

---

## [Editor Report · Acceptance letter]

8 Jul 2021

PONE-D-21-05956R1 

Maternal and neonatal canine cortisol measurement in multiple matrices during the perinatal period: a pilot study 

Dear Dr. Giordano:

I'm pleased to inform you that your manuscript has been deemed suitable for publication in PLOS ONE. Congratulations! Your manuscript is now with our production department. 

Kind regards, 

on behalf of

Dr. Benito Soto-Blanco 

Academic Editor

PLOS ONE